# Obesity-Associated NAFLD Coexists with a Chronic Inflammatory Kidney Condition That Is Partially Mitigated by Short-Term Oral Metformin

**DOI:** 10.3390/nu17132115

**Published:** 2025-06-26

**Authors:** Amod Sharma, Reza Hakkak, Neriman Gokden, Neelam Joshi, Nirmala Parajuli

**Affiliations:** 1Department of Pharmacology and Toxicology, University of Arkansas for Medical Sciences, Little Rock, AR 72205, USA; asharma2@uams.edu (A.S.); njoshi@uams.edu (N.J.); 2Department of Pediatrics, University of Arkansas for Medical Sciences, Little Rock, AR 72205, USA; rhakkak@uams.edu; 3Department of Dietetics and Nutrition, University of Arkansas for Medical Sciences, Little Rock, AR 72205, USA; 4Arkansas Children’s Research Institute, Little Rock, AR 72202, USA; 5Department of Pathology, University of Arkansas for Medical Sciences, Little Rock, AR 72205, USA; gokdenneriman@uams.edu

**Keywords:** obesity-NAFLD, kidney injury, inflammation, oxidative stress, metformin

## Abstract

**Background/Objectives:** Chronic kidney disease (CKD) is twice as prevalent in individuals with obesity-associated non-alcoholic fatty liver disease (Ob-NAFLD), highlighting the need to determine the link and mechanisms of kidney injury as well as explore therapies. Metformin, a first-line treatment for type 2 diabetes, shows promise in managing NAFLD, but its renal benefits in Ob-NAFLD remain unclear. This study investigates the impact of Ob-NAFLD on kidney injury and assesses the potential protective effects of metformin. **Methods:** Five-week-old female Zucker rats (obese fa/fa and lean Fa/Fa) were fed an AIN-93G diet for 8 weeks to induce Ob-NAFLD, then fed the diet with Metformin for 10 weeks. Kidneys were collected for histopathological and biochemical analyses. **Results:** Histopathological studies showed increased tubular injury, mesangial matrix expansion, and fibrosis in kidneys with Ob-NAFLD compared to lean control (LC) rats. Immunohistochemistry further revealed an elevated macrophage and neutrophil infiltration and increased levels of nitrotyrosine and p22phox in Ob-NAFLD kidneys. Furthermore, Ob-NAFLD rat kidneys showed upregulation of TNF-α and CCL2 genes and increased levels of caspase-3 (total and cleaved). Interestingly, metformin treatment significantly decreased TNF-α mRNA and blunted nitrotyrosine levels, and modestly reduced immune cell infiltration in Ob-NAFLD. **Conclusions:** These findings indicate that Ob-NAFLD promotes CKD as evidenced by tubular injury, oxidative stress, inflammation, and fibrosis. While short-term metformin treatment showed anti-oxidative and anti-inflammatory effects in Ob-NAFLD, its impact on structural kidney damage was limited, highlighting the need for longer treatment or alternative therapeutics such as oxidant scavengers and anti-inflammatory drugs to effectively mitigate renal pathologies.

## 1. Introduction

Chronic kidney disease (CKD) is a major global health concern, affecting an estimated 13.4% of the population and contributing significantly to morbidity and premature mortality worldwide [1]. Among the modifiable risk factors, obesity has emerged as a key contributor to CKD, primarily through its association with metabolic disorders such as type 2 diabetes, hypertension, and non-alcoholic fatty liver disease (NAFLD) [2,3,4]. Notably, the coexistence of obesity and NAFLD (Ob-NAFLD) appears to amplify the risk of renal complications [5]. When these two conditions coincide, the likelihood of developing CKD increases markedly. Epidemiological evidence indicates that individuals with Ob-NAFLD are 20–50% more likely to develop CKD compared to those without NAFLD, even after adjusting for conventional metabolic risk factors [6,7]. This suggests that Ob-NAFLD represents a distinct and potent risk factor contributing to kidney injury.

While this clinical association is well-documented, a clear link between Ob-NAFLD and kidney injury must be established. Similarly, the mechanisms exploring Ob-NAFLD-associated kidney injury remain incompletely understood. Ob-NAFLD is characterized by hepatic lipid accumulation, lipotoxicity, and hepatic inflammation, all of which elevate systemic levels of pro-inflammatory cytokines such as tumor necrosis factor-α (TNF-α) and monocyte chemoattractant protein (MCP-1) [8,9,10,11,12]. These circulating mediators could contribute to systemic low-grade inflammation and metabolic stress that may adversely affect distant organs, including the kidneys. Supporting this hypothesis, animals fed with a high-fat diet showed increased triglyceride and cholesterol contents in the kidneys and activated lipogenic pathways leading to glomerular and tubular injury [13]. Additionally, exposure of renal tubular cells to lipid results in oxidative stress, mitochondrial dysfunction, and apoptosis, all of which drive tubulointerstitial fibrosis and progressive decline in renal function [14].

Despite increasing recognition of the interorgan crosstalk between liver and kidney, relatively few preclinical studies have systematically investigated the renal consequences of Ob-NAFLD or tested interventions that target both hepatic and renal injury. Among the potential therapeutic candidates, metformin, a first-line treatment for type 2 diabetes, has shown beneficial effects beyond glycemic control. It improves hepatic insulin sensitivity, reduces steatosis, and suppresses systemic inflammation [15,16]. In addition to these hepatic benefits, emerging clinical data further suggest that metformin may offer reno-protective benefits. Observational and clinical studies have linked its use to a lower risk of CKD progression and all-cause mortality among individuals with type 2 diabetes [17,18], raising the possibility that metformin could also mitigate renal injury in the context of Ob-NAFLD.

Given the rising global prevalence of Ob-NAFLD and its potential role in promoting kidney pathophysiology, there is an urgent need to clarify the mechanisms of kidney injury in this setting and to evaluate potential therapeutic strategies that address both liver and kidney pathology. This study addresses this gap, in part, by investigating the extent of kidney injury, inflammation, and oxidative stress in a rat model of diet-induced Ob-NAFLD, and by assessing the potential reno-protective effects of short-term metformin treatment.

## 2. Materials and Methods

### 2.1. Ethics Statement

The animal care protocol and procedures in the study were approved by the University of Arkansas for Medical Sciences and Arkansas Children’s Research Institute Institutional Animal Care and Use Committee (Protocol code no. 3968; approved on 20 December 2019) and followed the guidelines of the United States Department of Agriculture (USDA, Washington, DC, USA) Animal Welfare Act.

### 2.2. Experimental Design

The experimental design and kidney tissue samples were adopted from prior research [15]. Briefly, five-week-old female Zucker rats (obese fa/fa, N = 6 and lean Fa/Fa, N = 6) were purchased from Envigo and housed individually with free access to water. All rats were fed an AIN-93 G diet from Envigo for 8 weeks to induce obesity and NAFLD (Ob-NAFLD) [15,19,20]. After this period, the rats were divided into four groups (n = 6): Lean Control (LC), Lean + Metformin (L + M), Ob-NAFLD, and Ob-NAFLD + Metformin (Ob-NAFLD + M). Metformin was administered for 10 weeks by mixing 1 g of metformin per kilogram of food. At the end of the treatment, the rats were sacrificed using 30% CO_2_ prior to decapitation. Kidney tissues were collected for histopathological and biochemical analysis.

### 2.3. Histology Analysis

Paraffin-embedded tissue blocks were sectioned at 4 µm thickness and were mounted on a glass slide and deparaffinized with xylene and a series of graded ethanol washes [21]. The rehydrated tissue sections were then stained with Periodic Acid–Schiff (PAS) and Masson’s Trichrome stain. The stained sections were examined under a light microscope (Nikon Eclipse), and a semi-quantitative scoring system was used to assess tubular injury, mesangial matrix expansion, and fibrosis. The scoring scale ranged from 0 (no changes), 1 (mild, affecting 1–25%), 2 (moderate, 26–50%), to 3 (severe, greater than 50%). A licensed pathologist carried out histopathological evaluation of kidney injury in a blinded fashion.

### 2.4. Immunohistochemistry Analysis

For immunohistochemistry, paraffin-embedded kidney tissue sections (4 µm thickness) were deparaffinized with xylene and a series of graded ethanol washes [22]. The rehydrated kidney sections underwent antigen retrieval via heating in sodium citrate buffer (pH 6.0) followed by quenching with BLOXALL™ Endogenous Peroxidase Solution (Vector Laboratories, Newark, NJ, USA). The slides were incubated with blocking solution (3% BSA and 0.5% non-fat dry milk in TBS) for 20 min at room temperature followed by incubation with primary antibodies overnight at 4 °C. The primary antibodies (Table 1) used were anti-CD68 (Abcam, Cambridge, UK), neutrophil elastase (Cell Signaling, Danvers, MA, USA), P22phox (Cell Signaling, Danvers, MA, USA), and anti-nitrotyrosine (Millipore, Burlington, VT, USA). Immunoreactivity was detected with ImmPRESS^®^ Polymer Detection Kit and reagent (Vector Laboratories, Newark, NJ, USA; MP-7451). All images were taken on a Nikon Eclipse Ni microscope with Nikon Elements software (BR 6.02.01) [23].

### 2.5. Western Blot Analysis

SDS-PAGE Western blotting was performed using rat kidney homogenates [24]. RIPA lysates (25 µg) per sample were separated on a 4–12% gradient gel (Thermo Fischer Scientific, Waltham, MA, USA), and proteins were transferred to a PVDF membrane for immunodetection (Thermo Fischer Scientific, Waltham, MA, USA). The PVDF membrane was incubated with the primary antibodies (Table 1), followed by horseradish peroxidase-conjugated secondary antibodies. β-actin was used as a loading control. The resulting signals were detected with an enhanced chemiluminescence kit (Thermo Fischer Scientific, Waltham, MA, USA; 34580), and iBright ™ CL 1500 Imaging System (Thermo Fischer Scientific, Waltham, USA). AlphaEase FC software (3.1.2) was used for densitometry and proteins normalized to β-actin.

### 2.6. Quantitative Real-Time PCR

An RNEasy kit (Qiagen, Aarhus, Denmark) was used to isolate total RNA from kidney tissue, and Superscript III (Invitrogen, Carlsbad, CA, USA) was used for reverse transcription of mRNA [23]. RT-PCR was carried out using the PowerUp SYBR Green Master Mix (Applied Biosystems, Waltham, MA, USA) and the PCR was run for 40 cycles with the following conditions: 95 °C for 15 s, 58 °C for 10 s, and 60 °C for 50 s. Amplification of target genes was normalized to actin amplification and to the levels of control using the ΔΔCt method [25]. The specific primer sequences used (RealTimePrimers.com) were as follows: chemokine (C-C motif) ligand 2 (Ccl2), forward 5′-TTGTCACCAAGCTCAAGAGA-3′ and reverse 5′-GGTTGTGGAAAAGAGAGTGG-3′; TNF-α, forward 5′-CCCATTACTCTGACCCCTTT-3′ and reverse 5′-TGAGCATCGTAGTTGTTGGA-3′; CXCR2 forward 5′-CAG AGA CTT GGG AGC CAC TC-3′ and reverse, 5′-TCA GCA AAG TCA CCA GAA CG-3′ and actin forward 5′-CAC ACT GTG CCC ATC TAT GA-3′ and reverse, 5′-CCG ATA GTG ATG ACC TGA CC-3.

### 2.7. Statistical Analysis

Kidneys isolated from the rats (n = 6 per group), as published before [15], were considered in this study. Statistical analysis was performed using GraphPad Prism (Version 10), and the data are presented as mean ± SD. The Shapiro–Wilk normality test was performed to verify normal distribution of the data. We performed a two-way ANOVA for comparisons among multiple groups. Tukey’s post hoc analysis (alpha = 0.05) was performed when normal distribution assumptions were met. For the skewed data, the Mann–Whitney U test was performed to compare the difference between the two groups. Statistical significance was defined as a *p*-value < 0.05. Graphs were generated using GraphPad Prism (10.0.2).

## 3. Results

### 3.1. Ob-NAFLD Induces Tubular Injury, Glomerular Sclerosis, and Interstitial Fibrosis, While Short-Term Metformin Treatment Did Not Notably Ameliorate Kidney Pathology

Previous studies demonstrated that female Zucker rats (fa/fa) that are fed with AIN-93 G diet produce obesity when compared to the female Zucker rats (Fa/Fa) [15,19,20]. The mean ± SD body weight (g) of lean and obese rats at the beginning of the experiment was as follows: lean, 98 ± 7 g and obese, 155 ± 17g [15]. After 10 weeks, the body weight of the rats was as follows: lean, 269 ± 26 g, and obese, 590 ± 41 [15]. The obese rats displayed significantly altered serum metabolites and developed NAFLD as indicated by increased liver weight and steatosis, when compared to the lean rats [15,19,20]. Furthermore, the obese rats show increased liver weight and impaired liver function when compared to the lean rats [15], suggesting the obesity-associated NAFLD (Ob-NAFLD). In this study, we sought to determine if Ob-NAFLD has any consequences on kidney injury. To investigate this, we used the same rats as described in [15,19,20]. Rat kidneys were isolated and processed for biochemical and histopathological analyses. We assessed kidney histopathology using Periodic Acid–Schiff and Masson’s Trichrome staining in the formalin-fixed paraffin-embedded kidney sections. The LC rats showed no signs of tubular (Figure 1Aa,B) or glomerular injury (Figure 1Ae,C), and the rats displayed no signs of renal fibrosis (Figure 1Ai,D). In contrast, Ob-NAFLD rats showed clear signs of kidney damage, including tubular injury (Figure 1Ac,B), mesangial matrix expansion (Figure 1Ag,C), and interstitial fibrosis (Figure 1Ak,D), when compared to LC rats. Pathological scoring showed that 20% of the Ob-NAFLD rats exhibited mild tubular injury, while 80% had moderate injury (Figure 1B). These results suggest that Ob-NAFLD contributes to a significant histopathological change in rat kidneys.

Previous studies utilizing the Ob-NAFLD rodent model demonstrated that the short-term metformin treatment (10 weeks) did not significantly affect body weight (lean rats: control, 269 ± 26 g; plus metformin, 278 ± 17 g; obese rats: obese, 590 ± 41, plus metformin, 573 ± 48 g) [15]. Although the metformin treatment did not alter the serum liver enzymes in Ob-NAFLD rats, it partially ameliorated liver histological changes (steatosis score) and improved serum metabolic parameters [15,19,20]. These findings underscore metformin’s therapeutic potential in partially managing liver damage, yet its impact on other organs affected by Ob-NAFLD, such as the kidneys, remains less explored. In this study, we tested the effects of oral metformin on kidney histopathology in lean and Ob-NAFLD rats. Unexpectedly, metformin-treated lean rats showed tubular injury when compared to lean controls (Figure 1Ab,B), but these rats displayed normal glomeruli (Figure 1Af,C) and no signs of fibrosis in kidneys (Figure 1Aj,D). Surprisingly, all metformin-treated Ob-NAFLD rats displayed moderate tubular injury (Figure 1Ad,B), mesangial matrix expansion (Figure 1Ah,C), and interstitial fibrosis (Figure 1Al,D), similar to the levels of Ob-NAFLD rats, suggesting that short-term oral metformin (low dose) is not effective in ameliorating renal injury in Ob-NAFLD rats.

### 3.2. Ob-NAFLD Increases Nitrotyrosine Levels in the Kidneys, Which Were Reduced by Short-Term Oral Metformin

To assess whether Ob-NAFLD induces oxidative stress in the kidneys, we examined the levels of nitrotyrosine (a marker of ROS) in the kidney sections. The immunohistochemistry showed a basal level of nitrotyrosine staining in LC rat kidneys (Figure 2Aa). Similarly, metformin-treated LC rats (L + M) showed similar levels of nitrotyrosine in kidneys compared to the LC group (Figure 2Ab). A modest increase in nitrotyrosine was observed in Ob-NAFLD kidneys compared to the LC with or without metformin groups (Figure 2Ac). Interestingly, low-dose oral metformin treatment reduced nitrotyrosine levels in Ob-NAFLD kidneys (Figure 2Ad). These findings suggest that while oxidative stress is a contributing factor and metformin blunts it, the oxidants are not the sole driver of kidney injury in Ob-NAFLD rats.

### 3.3. Ob-NAFLD Increases Macrophage and Neutrophil Infiltration in the Kidneys, Which Is Mildly Reduced by Short-Term Metformin Treatment

Given the renal oxidative stress (Figure 2), which may induce other mechanisms such as inflammation, we focused on studying the presence of immune infiltrates within the kidneys after Ob-NAFLD. We performed immunohistochemistry using antibodies against CD68 (macrophages) and elastase (neutrophils). Our results showed sporadic or no CD68+ (Figure 3Aa) and elastase+ cells (Figure 3Ba) within the kidney parenchyma of LC rats, and similar observations were noted for LC rats treated with metformin (Figure 3Ab,Bb). However, increased infiltration of CD68-positive cells (Figure 3Ac,C) and neutrophils (Figure 3Bc,D) was observed in the Ob-NAFLD rat kidneys compared to the LC or L + M groups. A modest reduction in infiltration was seen in the kidneys after short-term metformin treatment in Ob-NAFLD rats (Ob-NAFLD + M) (Figure 3Ad,Bd,C,D). Because the infiltrated immune cells can generate ROS, we evaluated the levels of p22phox, a subunit of the NADPH oxidase complex and a contributor of ROS production [26]. As in Figure 3, control groups LC (Figure 4Aa) and L + M (Figure 4Ab) showed a basal level of p22phox, mainly localized in the infiltrating cell compartment. Compared to these control groups, elevated levels of p22phox were observed in the infiltrating cells within the kidneys of the Ob-NAFLD group (Figure 4Ac). Unexpectedly, rats treated with low-dose oral metformin did not reduce p22phox levels in Ob-NAFLD + M kidneys (Figure 4Ad). These findings suggest that p22phox may not be a contributor to the observed oxidative stress in Ob-NAFLD rats.

### 3.4. Ob-NAFLD Selectively Modulates Renal Antioxidant Enzymes, and Metformin Differentially Affects GST-P, SOD1, and SOD2 Levels

Given the evidence of modest oxidative stress in the kidneys of Ob-NAFLD rats (Figure 2), we assessed the expression of key antioxidant enzymes. Using Western blotting, we evaluated glutathione-S-transferase-P (GST-P), a protein involved in detoxification and redox homeostasis [27,28], copper-zinc superoxide dismutase (Cu/Zn-SOD or SOD1), a cytosolic antioxidant [29], and manganese superoxide dismutase (Mn-SOD or SOD2), a mitochondrial antioxidant [30] in kidney homogenates. GST-P levels were significantly elevated in the kidneys of Ob-NAFLD rats compared to the LC group (Figure 5A,B), suggesting an adaptive response to oxidative stress. GST-P is a phase II detoxification enzyme that catalyzes the conjugation of glutathione to reactive electrophilic compounds, thereby playing a critical role in protecting cells from oxidative and xenobiotic insults [31]. In contrast, the expression levels of Cu/Zn-SOD and Mn-SOD, both of which catalyze the dismutation of superoxide radicals into hydrogen peroxide and molecular oxygen [32], remained unchanged between the LC and Ob-NAFLD groups (Figure 5A,C,D). This suggests that protein levels of these classical antioxidant defenses were not significantly modulated in the kidneys of the Ob-NAFLD rats. To evaluate the effects of metformin on renal antioxidant defense mechanisms, we considered lean and Ob-NAFLD rats orally fed with metformin. As expected, we did not observe any changes in the protein levels of GST-P, Cu/ZnSOD, and MnSOD in the kidneys of lean rats treated with metformin when compared to lean rats (Appendix A). Interestingly, metformin treatment significantly increased protein levels of Cu/ZnSOD and decreased MnSOD levels in kidneys but did not alter GST-P levels in Ob-NAFLD + M rats compared to L + M or Ob-NAFLD groups (Figure 5E–H). This differential modulation of antioxidant enzymes suggests metformin’s potential effect on cellular redox signaling via a shift in antioxidant response toward cytosolic defense through Cu/ZnSOD.

### 3.5. Ob-NAFLD Upregulates the Gene Expression of Pro-Inflammatory Markers TNF-α, MCP-1, and CXCR2 in the Kidney, While Metformin Treatment Specifically Reduces TNF-α Expression

To further explore the mechanisms underlying macrophage and neutrophil recruitment in the kidneys following Ob-NAFLD, we analyzed gene expression of key cytokines and chemokines. TNF-α is a central cytokine involved in promoting inflammation and immune cell activation [33]. C-C motif chemokine ligand 2 (Ccl2), plays a crucial role in recruiting monocytes/macrophages to sites of inflammation [34,35]. Meanwhile, C-X-C chemokine receptor 2 (CXCR2) primarily mediates neutrophil chemotaxis in response to its ligands [30]. Notably, Ccl2 and CXCR2 act through distinct but complementary pathways to facilitate immune cell infiltration during renal injury [36,37]. We evaluated the levels of gene expression for TNF-α, Ccl2, and CXCR2 in rat kidneys after Ob-NAFLD with or without metformin treatment. Compared to LC group, Ob-NAFLD rat kidneys exhibited significantly increased mRNA expression of TNF-α, Ccl2, and CXCR2 (Figure 6 A–C), suggesting their potential role in promoting kidney inflammation and immune cell recruitment. Metformin treatment in lean rats (L + M) did not alter the expression levels of TNF-α (Figure 6A) and Ccl2 (Figure 6B), but it modestly increased CXCR2 gene expression in lean rats (Figure 6C, *p* = 0.07). Interestingly, metformin treatment in Ob-NAFLD rats (Ob-NAFLD + M) led to a significant reduction in TNF-α mRNA levels compared to the untreated Ob-NAFLD group (Figure 6A), whereas Ccl2 and CXCR2 levels remained unchanged (Figure 6 B,C). These findings suggest that metformin may attenuate inflammation in Ob-NAFLD kidneys by targeting specific inflammatory pathways, particularly those involving TNF-α.

### 3.6. Ob-NAFLD Enhances Apoptosis in Kidneys; Short-Term Oral Metformin Offers Modest Modulation

To assess whether Ob-NAFLD-induced kidney damage involves the activation of apoptotic pathways, we focused on caspase-3, a key executioner protease in the intrinsic apoptosis cascade. Caspase-3 is synthesized as an inactive proenzyme (full-length form) and becomes activated through cleavage in response to cellular stress, ultimately driving the execution phase of programmed cell death [38]. Given its central role in apoptosis, we analyzed the expression of both caspase-3 and cleaved caspase-3 in renal tissues using Western blotting to investigate its involvement in Ob-NAFLD-associated tubular injury (Figure 7). The protein levels of both forms were significantly elevated in the kidneys of Ob-NAFLD rats compared to LC rats (Figure 7A–C), indicating enhanced apoptotic activity. As expected, we did not observe any changes in the protein levels of total and cleaved caspase-3 in the kidneys of lean rats treated with metformin when compared to lean control (Appendix A). While the metformin-treated Ob-NAFLD group (Ob-NAFLD + M) resulted in a modest reduction in these apoptotic markers, the changes did not reach statistical significance (Figure 7D–F). These findings suggest that the apoptotic pathway contributes to Ob-NAFLD-associated kidney injury, and that short-term metformin therapy may offer limited protection against this type of cell death.

## 4. Discussion

The present study demonstrates that Ob-NAFLD leads to significant renal pathology, including tubular injury, glomerular sclerosis, mesangial matrix expansion, and interstitial fibrosis, hallmarks of chronic kidney disease. Although NAFLD is primarily considered a hepatic disorder, a growing body of evidence indicates that its metabolic and inflammatory consequences are systemic, extending well beyond the liver. Chronic low-grade inflammation, insulin resistance, and lipotoxicity associated with NAFLD are thought to contribute to the onset and progression of renal injury [2,39]. Consistent with this, our findings reveal that Ob-NAFLD-induced structural kidney damage is accompanied by increased immune cell infiltration, elevated oxidative stress, upregulated pro-inflammatory cytokines, and enhanced apoptosis-highlighting a multifactorial mechanism of injury. Although short-term metformin treatment modestly reduced selected markers of inflammation (TNF-α) and oxidative stress, it did not significantly alleviate the histological or molecular features of kidney injury, suggesting limited therapeutic efficacy under the current experimental conditions and dosing regimen. Histological analyses revealed substantial damage in both the tubular and glomerular compartments, characterized by moderate tubular injury and FSGS (focal segmental glomerulosclerosis) in the majority of Ob-NAFLD rats. Interstitial fibrosis, a hallmark of chronic kidney injury, was also present. These findings align with prior studies that have linked metabolic syndrome and NAFLD to CKD, potentially through mechanisms involving systemic inflammation, lipid toxicity, and alterations in renal hemodynamics [40,41]. The evidence supports the notion that Ob-NAFLD contributes to kidney injury via a complex interplay of systemic metabolic disturbances that extend beyond liver dysfunction.

This study showed infiltrating CD68+ and neutrophil elastase+ cells as a key hallmark of renal inflammation following Ob-NAFLD-associated renal injury. These immune cells play a pivotal role in a disease pathology by releasing pro-inflammatory cytokines, reactive oxygen species, and proteolytic enzymes that exacerbate tubular injury and interstitial fibrosis [42,43,44]. Tissue injury within the kidney leads to upregulated expression of chemokines such as Ccl2, a potent chemoattractant that facilitates the recruitment of monocytes/macrophages and neutrophils to sites of inflammation [45,46]. The presence of chemokine receptors like CXCR2 on these immune cells further allows them to respond to ligands like Ccl2, guiding their migration to the damaged tissue and amplifying the local inflammatory response [47]. This inflammatory milieu is further intensified by elevated TNF-α, a central mediator of renal inflammation in metabolic conditions [36,48]. In our study, metformin treatment significantly reduced TNF-α gene expression and modestly decreased macrophage and neutrophil infiltration, suggesting that it may confer partial anti-inflammatory effects by dampening key components of the inflammatory signaling cascade.

Our findings also demonstrate that oxidative stress plays a pivotal role in the pathophysiology of Ob-NAFLD-associated kidney injury. Notably, elevated levels of nitrotyrosine and p22phox in Ob-NAFLD kidneys suggest enhanced ROS production, likely driven by immune cell activation and upregulation of the NADPH oxidase complex [49,50]. This raises the question: is oxidative stress the primary instigator of renal tissue injury, or is it a consequence of infiltrating immune cells? While both possibilities are likely interconnected, the selective upregulation of GST-P indicates an adaptive cellular response to increased oxidative burden, suggesting that oxidative stress may be an early contributor to kidney damage. In contrast, the expression of classical antioxidant enzymes-SOD1 and SOD2 levels remained unchanged during Ob-NAFLD, suggesting limited engagement of canonical ROS-detoxifying pathways.

Metformin treatment significantly reduced nitrotyrosine levels, consistent with its known antioxidative properties [51,52]. However, this reduction did not lead to meaningful histological improvement, implying that oxidative stress is only one of several contributing factors in Ob-NAFLD-induced renal injury. Interestingly, metformin also led to differential regulation of SOD isoforms, with increased SOD1 and decreased SOD2 expression, suggesting a compartment-specific modulation of antioxidant defenses. Despite these molecular changes, metformin failed to produce a substantial shift in the overall antioxidant enzyme profile, likely due to limitations in treatment duration or dosage. Collectively, these results emphasize the complexity of oxidative mechanisms in Ob-NAFLD-related kidney damage and underscore the need for comprehensive therapeutic strategies that address multiple pathological pathways.

Increased apoptosis, as indicated by elevated levels of cleaved caspase-3 in Ob-NAFLD kidneys, is another important mechanism contributing to tubular injury. While metformin treatment led to a slight reduction in apoptotic markers, the changes were not statistically significant. This raises several possibilities worth considering. One likely reason could be the short duration of treatment, which might not have been long enough to reverse well-established apoptotic pathways. Another possibility is that the dose of metformin used may have been too low to exert a strong effect, especially in the context of chronic kidney injury. Moreover, variation in drug exposure among the rats might have contributed to the lack of a clear effect. Since the animals had ad libitum access to food, and metformin was provided through the diet, it is possible that not all rats consumed the same amount. This uneven intake could have resulted in some animals receiving subtherapeutic doses, thereby introducing intra-group variability and weakening the overall treatment response. Altogether, these factors highlight the complexity of kidney injury in the setting of Ob-NAFLD, where multiple overlapping mechanisms-such as apoptosis, oxidative stress, and inflammation-work in tandem. The findings also emphasize the need for longer studies, optimized dosing strategies, and perhaps more controlled drug delivery methods to better assess the potential of metformin in protecting against renal damage in this model.

One of the limitations of this study is the exclusive use of female Zucker rats. While this may raise questions regarding sex-specific responses, existing evidence suggests that in the Ob-NAFLD model, both male and female obese Zucker rats develop obesity and hepatic steatosis at comparable rates, with no significant sex-related differences in the progression of metabolic or histological features [19,53]. Therefore, the choice of female animals is unlikely to have introduced a substantial bias in the evolution of body weight and liver steatosis, including the key endpoints of the published studies [15]. Nonetheless, future studies could benefit from including both sexes to fully rule out any subtle, sex-specific variations in renal response to therapeutic interventions, particularly to the CKD condition. Similarly, our study design did not include the evaluation of blood glucose levels. Another limitation lies in the lack of functional renal assessments. While the study focused primarily on histological changes and molecular markers of kidney injury, we could not evaluate serological functional parameters of kidney health, which are critical for assessing therapeutic efficacy in a translational context. Functional biomarkers such as glomerular filtration rate, creatinine clearance, blood urea nitrogen, or proteinuria would provide valuable insights into how molecular and histological improvements translate into preserved or improved renal function. Similarly, the administration of metformin was performed via the oral route with ad libitum diet access, and the monitoring of food intake was not included in the experimental design, which is another significant limitation of this study.

## 5. Conclusions

This study demonstrates that kidney injury in the context of Ob-NAFLD is driven by a complex interplay of inflammation, oxidative stress, and apoptosis. While metformin treatment resulted in partial attenuation of select inflammatory and oxidative stress markers, it did not confer significant protection against renal structural damage or apoptosis over the short treatment period. These findings suggest that although metformin shows some promise, its renoprotective effects may be limited under short-term exposure and highlight the need for longer-duration studies, optimized dosing, and functional assessments to fully understand its therapeutic potential in Ob-NAFLD-associated kidney injury.

## Figures and Tables

**Figure 1 nutrients-17-02115-f001:**
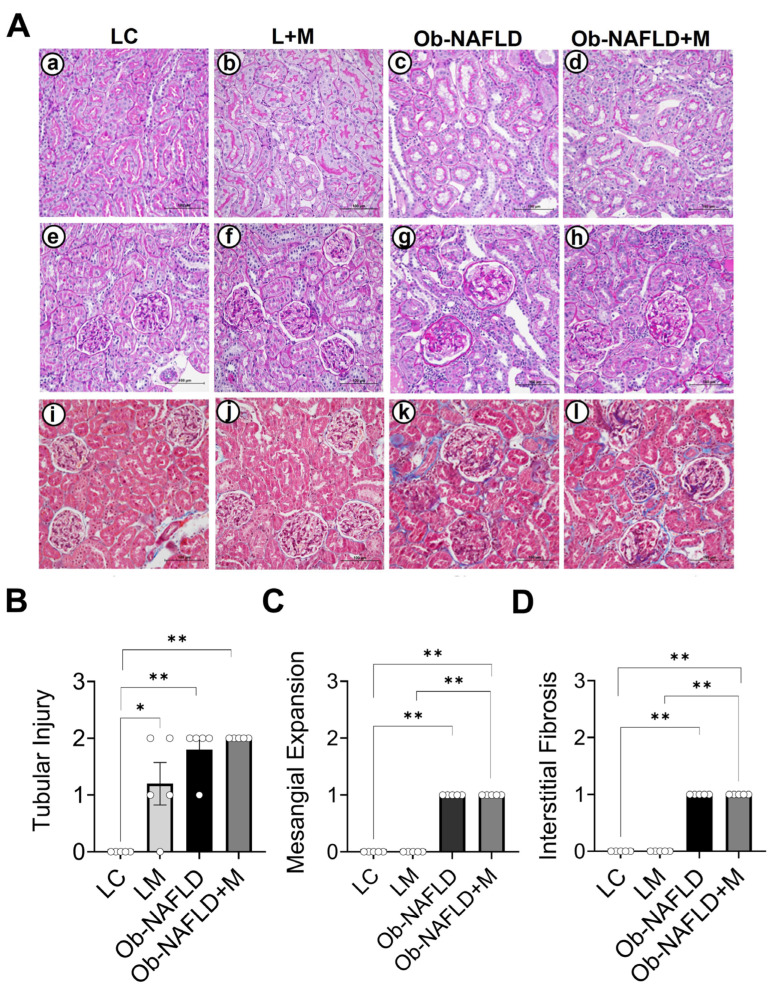
Effects of Ob-NAFLD and metformin treatment on rat kidney histopathology. Rat kidneys were isolated and processed for formalin-fixed paraffin-embedded sections. Three groups, namely lean control (LC), obesity associated NAFLD (Ob-NAFLD), and metformin treated Ob-NAFLD (Ob-NAFLD + M) were considered for histological evaluations. (**A**) Representative images of Periodic Acid–Schiff (PAS) (**a**–**h**) and Masson’s Trichrome-stained kidney sections (**i**–**l**). (**a**,**e**,**i**) LC rats showing normal renal histology with intact tubular structure (**a**), no mesangial expansion (**e**), and absence of fibrosis (**i**). (**b**,**f**,**j**) Metformin-treated lean rats exhibiting tubular injury (**b**), no mesangial matrix expansion (**f**), and absence of interstitial fibrosis (**j**). (**c**,**g**,**k**) Ob-NAFLD rats exhibiting evident kidney damage, including tubular injury (**c**), mesangial matrix expansion (**g**), and interstitial fibrosis (**k**). (**d**,**h**,**l**) Metformin-treated Ob-NAFLD rat kidneys showing pathological features similar to untreated Ob-NAFLD rats, including moderate tubular injury (**d**), mesangial expansion (**h**), and fibrosis (**l**), indicating a lack of renal protective effect from short-term metformin treatment. Scale bars = 100 μm. (**B**–**D**) Graphs showing pathological evaluation performed by a blinded renal pathologist. (**B**) Tubular injury scores, (**C**) mesangial matrix expansion score, and (**D**) interstitial fibrosis score across experimental groups. Data are expressed as mean ± SEM (n = 5 per group). Statistical significance between the two groups was determined by a non-parametric Mann–Whitney U test (* *p* < 0.05; ** *p* <0.01).

**Figure 2 nutrients-17-02115-f002:**
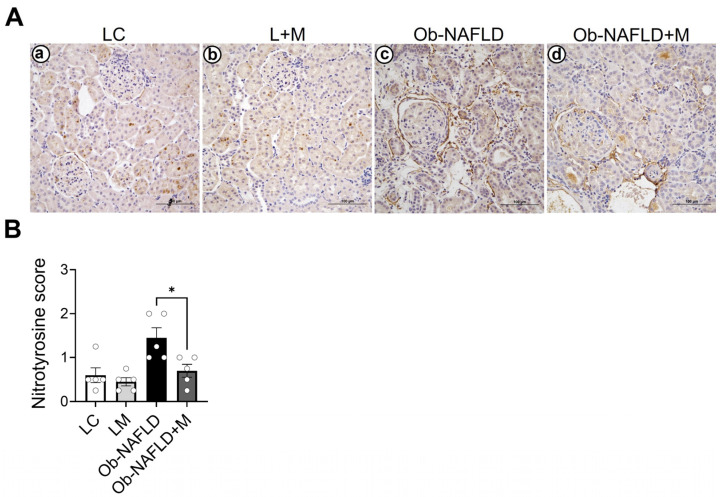
Evaluation of oxidative stress in the kidneys of Ob-NAFLD rats with or without metformin treatment. Rat kidney sections from lean control (LC), lean treated with metformin (L + M), obesity associated NAFLD (Ob-NAFLD), and metformin-treated Ob-NAFLD (Ob-NAFLD + M) groups were employed for immunohistochemistry of nitrotyrosine (a marker of reactive oxygen species). (**A**) Representative immunohistochemical images of nitrotyrosine staining (**a**–**d**). Scale bars = 100 μm. (**B**) Graph showing semi-quantitative analysis of nitrotyrosine levels across experimental groups. Data are expressed as mean ± SEM (n = 5 per group). Statistical significance was determined by two-way ANOVA (* *p* < 0.05).

**Figure 3 nutrients-17-02115-f003:**
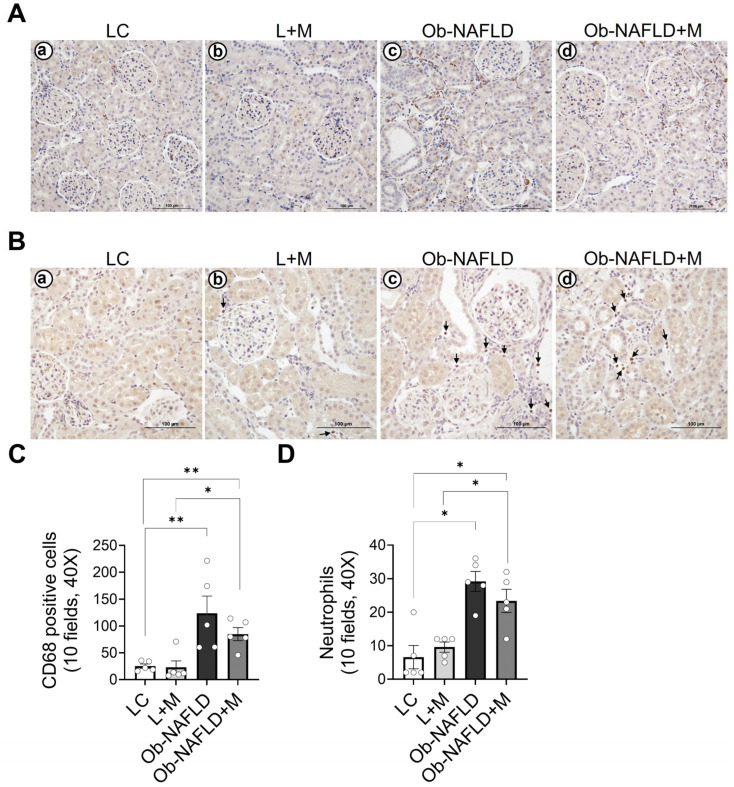
Immune cell infiltration in the kidneys of Ob-NAFLD rats with or without metformin treatment. Rat kidney sections from lean control (LC), lean treated with metformin (L + M), obesity associated NAFLD (Ob-NAFLD), and metformin-treated Ob-NAFLD (Ob-NAFLD = M) groups were employed for immunohistochemistry of CD68, a macrophage marker, and neutrophil elastase, a neutrophil marker. (**A**) Representative immunohistochemistry images showing CD68+ cells in rat kidneys of the following: (**a**) LC; (**b**) L + M; (**c**) Ob-NAFLD; and (**d**) Ob-NAFLD + M groups. Scale bars = 100 μm. (**B**) Representative immunohistochemistry images for neutrophil elastase+ cells in rat kidneys of the following: (**a**) LC; (**b**) L + M; (**c**) Ob-NAFLD; and (**d**) Ob-NAFLD + M groups. Scale bars = 100 μm. (**C**) Graph showing quantification of CD68+ cells within 10 microscopic fields (40×) across experimental groups. (**D**) Graph showing quantification of neutrophil elastase+ cells within 10 microscopic fields (40×) across experimental groups. Data are expressed as mean ± SEM (n = 5 per group). Statistical significance between the two groups was determined by a non-parametric Mann–Whitney U test (* *p* < 0.05; ** *p* < 0.01).

**Figure 4 nutrients-17-02115-f004:**
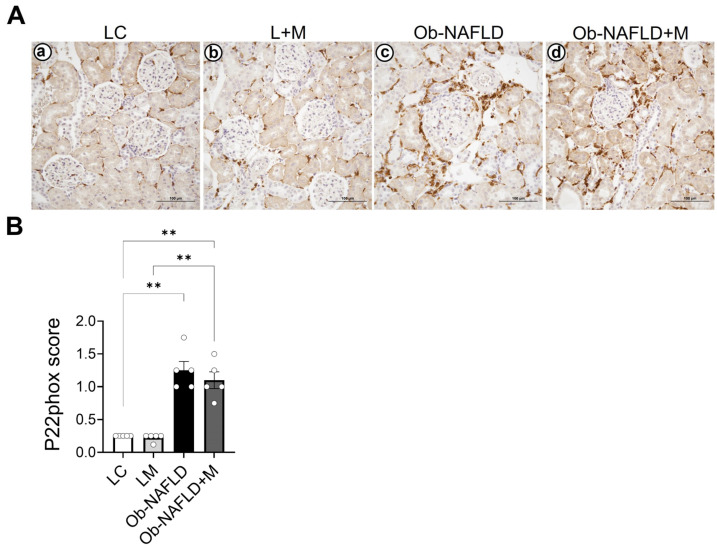
Expression of p22phox, a subunit of NADPH oxidase, in kidney tissues of Ob-NAFLD rats. Rat kidney sections from lean control (LC), lean treated with metformin (L + M), obesity associated NAFLD (Ob-NAFLD), and metformin-treated Ob-NAFLD (Ob-NAFLD = M) groups were employed for immunohistochemistry of p22phox, a subunit of NAPDH complex and a marker of reactive oxygen species production. (**A**) Representative immunohistochemistry images showing p22phox+ cells in rat kidneys of the following: (**a**) LC; (**b**) L + M; (**c**) Ob-NAFLD; and (**d**) Ob-NAFLD + M groups. Scale bars = 100 μm. (**B**) Graph showing quantification of p22phox+ cells within 10 microscopic fields (40×) across experimental groups. Data are expressed as mean ± SEM (n = 5 per group). Statistical significance between the two groups was determined by a non-parametric Mann–Whitney U test (** *p* < 0.01).

**Figure 5 nutrients-17-02115-f005:**
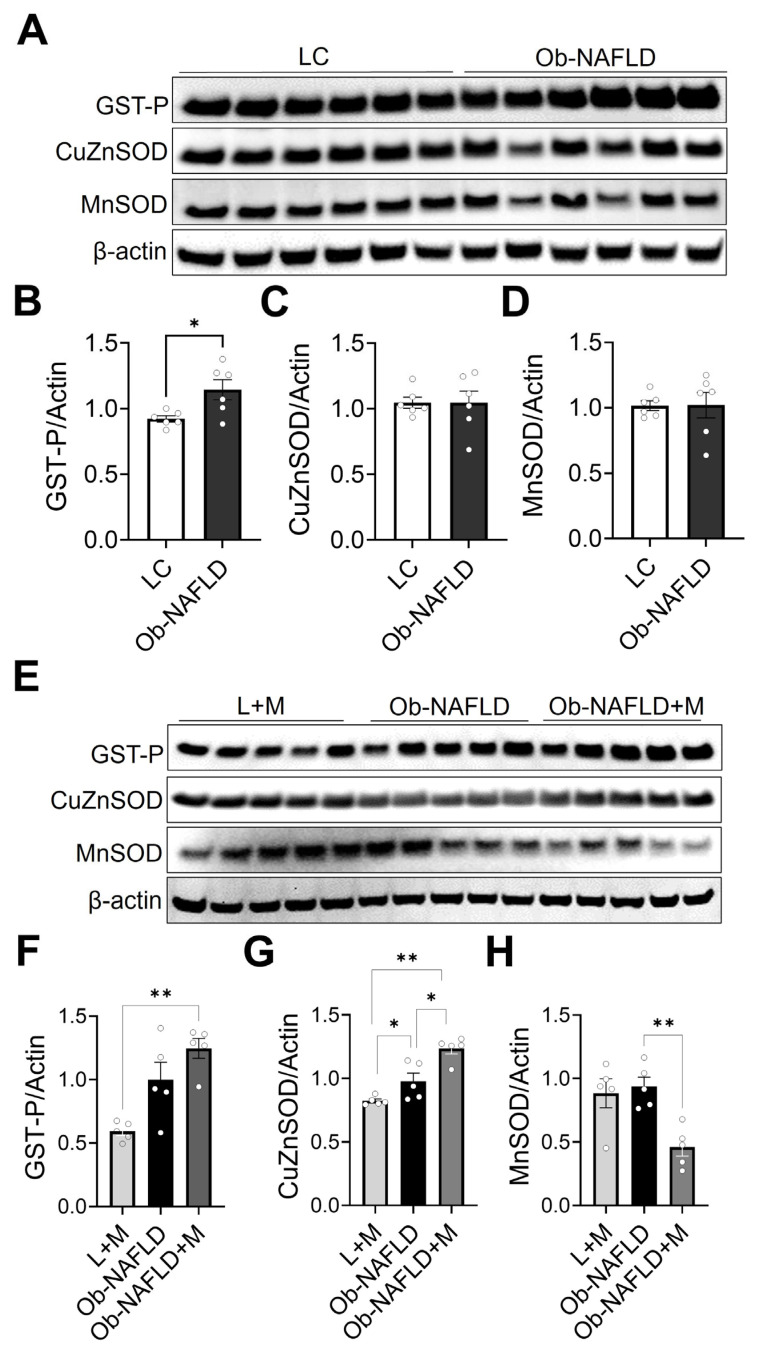
Antioxidant enzyme levels in the kidneys of Ob-NAFLD rats and the impact of metformin treatment. RIPA lysates were prepared from rat kidney homogenates followed by SDS-PAGE Western blotting. (**A**) Representative Western blot images of glutathione-S-transferase-P (GST-P), Cu/Zn-superoxide dismutase (Cu/Zn-SOD), and Mn-superoxide dismutase (Mn-SOD) in renal lysates from lean control (LC) and obesity associated NAFLD (Ob-NAFLD) groups. (B-D) Graphs showing densitometry analysis of protein bands of GST-P (**B**), CuZnSOD (**C**), and MnSOD (**D**) normalized to β-actin. (**E**) Representative Western blot images of glutathione-S-transferase-P (GST-P), Cu/Zn-superoxide dismutase (Cu/Zn-SOD), and Mn-superoxide dismutase (Mn-SOD) in renal lysates from lean treated with metformin (L + M), Ob-NAFLD, and metformin-treated Ob-NAFLD (Ob-NAFLD + M) groups. (F-H) Graphs showing densitometry analysis of protein bands of GST-P (**F**), CuZnSOD (**G**), and MnSOD (**H**) normalized to β-actin. Data are presented as mean ± SEM (n = 5 per group). Statistical significance between the two groups was determined by a non-parametric Mann–Whitney U test (* *p* < 0.05, ** *p* < 0.01).

**Figure 6 nutrients-17-02115-f006:**
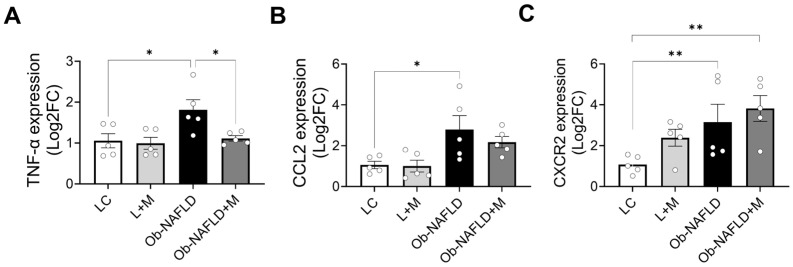
Expression of inflammatory cytokine and chemokine genes in the kidneys of Ob-NAFLD rats with or without metformin treatment. Kidney tissues from lean control (LC), lean treated with metformin (L + M), Ob-NAFLD, and Ob-NAFLD treated with metformin (Ob-NAFLD + M) groups were used for mRNA isolation followed by complementary DNA (cDNA) synthesis. (**A**–**C**) SYBR green PCR was used to quantify gene expression of tumor necrosis factor (TNF-α), a pro-inflammatory cytokine, chemokine ligand 2 (Ccl2), a chemoattractant for immune cell infiltration, and the CXC chemokine receptor 2 (CXCR2). Gene expression levels were normalized to β-actin and expressed as fold change relative to the LC group. Graphs showing quantitative real-time PCR analysis of TNF-α (**A**), Ccl2 (**B**), and CXCR2 (**C**) mRNA expression. Data are presented as mean ± SEM (n = 5 per group). Statistical analysis was performed using two-way ANOVA (* *p* < 0.05; ***p* < 0.01).

**Figure 7 nutrients-17-02115-f007:**
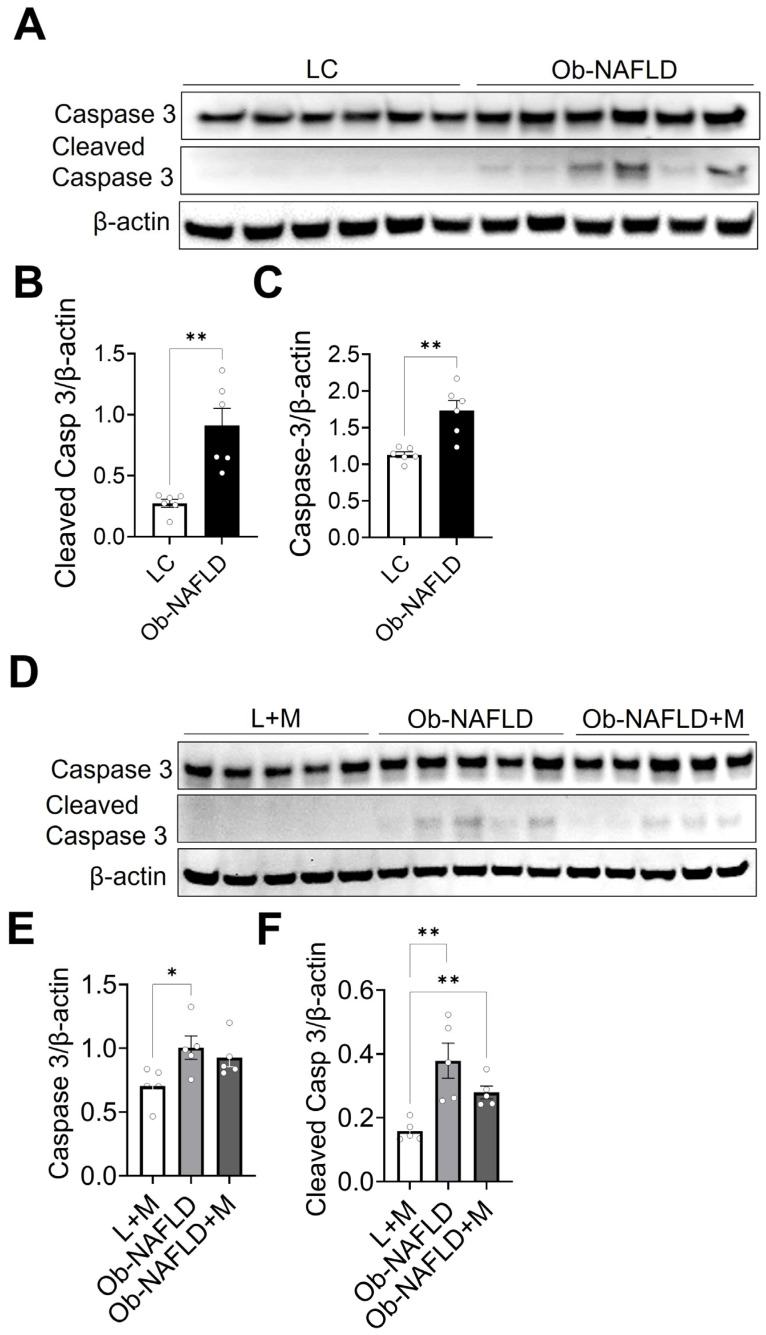
Effects of Ob-NAFLD on renal apoptosis and the impact of metformin treatment. RIPA lysates were prepared from rat kidney homogenates, followed by SDS-PAGE Western blotting. (**A**) Representative Western blot images of total caspase-3 and cleaved caspase-3 proteins in renal lysates from lean control (LC) and obesity associated NAFLD (Ob-NAFLD) groups. (**B**,**C**) Graphs showing densitometry analysis of protein bands of total caspase-3 (**B**) and cleaved caspase-3 proteins (**C**) normalized to β-actin. (**D**) Representative Western blot images of total caspase-3 and cleaved caspase-3 proteins in renal lysates from lean treated with metformin (L + M), Ob-NAFLD, and metformin-treated Ob-NAFLD (Ob-NAFLD + M) groups. (**E**,**F**) Graphs showing densitometry analysis of protein bands of total caspase-3 (**E**) and cleaved caspase-3 proteins (**F**) normalized to β-actin. Data are presented as mean ± SEM (n = 5 per group). Statistical significance between the two groups was determined by a non-parametric Mann–Whitney U test (* *p* < 0.05; ***p* < 0.01).

**Table 1 nutrients-17-02115-t001:** Primary antibodies used in this study.

Name	Company	Catalog No.	Host	Dilution
Cu/ZnSOD	MilliporeSigma	HPA001401	Rabbit	1:1000 (WB)
Ccl2	LSBio	C201039	Rabbit	1:1000 (WB)
MnSOD	MilliporeSigma	6984	Rabbit	1:1000 (WB)
GST-P	MBL	311	Rabbit	1:1000 (WB)
Caspase 3	Cell Signaling	14220	Rabbit	1:1000 (WB)
Cleaved Caspase 3	Cell Signaling	14220	Rabbit	1:1000 (WB)
Actin	Invitrogen	MA5-15739	Mouse	1:1000 (WB)
CD68	Abcam	ab125212	Rabbit	1:200 (IHC)
Nitrotyrosine	Millipore	06-284	Rabbit	1:2000 (IHC)
Neutrophil Elastase	Cell Signaling	44030	Rabbit	1:400 (IHC)
P22phox	Cell Signaling	37570	Rabbit	1:1000 (IHC)

## Data Availability

All data associated with this study are present in the paper. Data can be shared upon reasonable request to the corresponding author.

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
