# Peer review of "Obesity-Associated NAFLD Coexists with a Chronic Inflammatory Kidney Condition That Is Partially Mitigated by Short-Term Oral Metformin"

_nutrients, 2025, doi:10.3390/nu17132115_

Round 1
Reviewer 1 Report
Comments and Suggestions for Authors
The manuscript addresses a relevant topic and presents potentially valuable findings. It is also generally well-written, carefully prepared, and well discussed. However, I have several comments and suggestions for the authors to consider:
- Title: Since the authors do not clearly demonstrate a specific mechanism linking Ob-NAFLD to kidney disease, I believe the main contribution of the study lies in demonstrating the coexistence of Ob-NAFLD and CKD, along with the (albeit modest) beneficial effect of metformin. Therefore, I suggest revising the title to better reflect this key finding. For example: “Obesity-associated NAFLD coexists with a chronic inflammatory kidney condition that is partially mitigated by short-term oral metformin treatment.”
- Abstract, lines 19–21: The phrase "normal diet to induce Ob-NAFLD" may be confusing, as a "normal" diet, by definition, should not induce NAFLD. Although this is clarified later in the manuscript, the wording in the abstract may not be immediately clear to the reader. Consider removing the word "normal", or alternatively, specifying the actual name of the diet (e.g., AIN-93G) to enhance clarity.
- Given that the authors’ main hypothesis relies on the combined effect of obesity and NAFLD on the development of kidney disease, it is essential to clearly demonstrate that the animals used in this study were both obese and had NAFLD. While the use of a genetic obesity model supports the presence of obesity, the manuscript does not provide sufficient evidence that the animals developed NAFLD, neither it is specified in which degree or severity. It appears that these data may have been reported in previous publications. Using the same animals is, of course, entirely acceptable. However, in its current form, the manuscript does not clearly state whether the rats used in this study are the same as those described in the cited references. The authors should explicitly clarify this point in the text and provide the relevant references to confirm that the animals analyzed in the present study met both conditions.
- Closely related to the previous comment, I find that the manuscript lacks some important biometrical and biochemical data for the rats, which are particularly relevant to assess the degree of obesity and the severity of NAFLD the rats devoloped, such as initial and final body weight, food intake, glucose status, etc. If these data have already been published elsewhere, please clearly provide the corresponding references so that readers can consult them. Additionally, a brief summary of the main health characteristics of the rats could be helpful (at the authors’ discretion, if the references are provided).
- Along the same lines, in the Discussion, lines 429-434: The authors suggest that variability in metformin intake due to ad libitum feeding may have contributed to the modest or inconsistent treatment effects. While this is a plausible explanation, it underscores a methodological limitation that should be more directly acknowledged. As noted in my previous comment, the manuscript does not include any biometric data—such as body weight evolution or food intake—which are essential when drug administration depends on dietary consumption. Even with ad libitum access, monitoring food intake per cage and correlating it with individual body weight would allow for a rough estimation of drug exposure per animal. This is a key point that the authors should clarify, as it directly impacts the interpretation of metformin's effectiveness. Including such data, or explicitly addressing its absence as a limitation, would considerably strengthen the validity of the conclusions.
- I am not entirely sure about the statistical approach used. Why was a one-way ANOVA chosen? Which factor was considered for the analysis? It seems that there might actually be two relevant factors—rat strain and metformin treatment—so a two-way ANOVA could be more appropriate. Additionally, ANOVA (whether one-way or two-way) does not perform pairwise comparisons by itself. In the text, the authors state that t-tests were used for this purpose, but this is not reflected in the figure legends or results. Finally, although the answer is likely yes, did you verify whether the data meet the assumptions for parametric tests? This should be clearly specified in the methods section.
- I kindly suggest that the data mentioned in the manuscript but not shown be included as supplementary material. This would enhance transparency and allow readers to fully assess the study’s conclusions.
- Regarding the organization of the manuscript, I do not believe that placing all the figures under a single section is the best choice. It would be much clearer and more reader-friendly to position each figure as close as possible to its first citation in the text. In any case, the separate figure section should be removed to improve the overall flow and readability of the manuscript.
- Line 152: Please add the corresponding references and clearly specify whether these are the same rats. As mentioned earlier, this point is critical to support the authors' hypothesis.
- Minor textual issues:
- If I am correct, Table 1 lists only the primary antibodies used in the study, despite the title suggesting otherwise. The authors may want to revise the table title to accurately reflect its contents.
- Line 142: Please correct the capitalization of "Reverse" to "reverse."
- Lines 245-246: Please, revise this sentence: “We evaluated the levels of gene expression for TNF-α, Ccl2, and CXCR2 in rat kidneys after Ob-NAFL with or without metformin treatment.”
- Line 378: Please, define FSGS.
Author Response
Response to the Reviewers.
Comments from Reviewer 1 and Author’s Response
The manuscript addresses a relevant topic and presents potentially valuable findings. It is also generally well-written, carefully prepared, and well discussed. However, I have several comments and suggestions for the authors to consider:
1. Title: Since the authors do not clearly demonstrate a specific mechanism linking Ob-NAFLD to kidney disease, I believe the main contribution of the study lies in demonstrating the coexistence of Ob-NAFLD and CKD, along with the (albeit modest) beneficial effect of metformin. Therefore, I suggest revising the title to better reflect this key finding. For example: “Obesity-associated NAFLD coexists with a chronic inflammatory kidney condition that is partially mitigated by short-term oral metformin treatment.”
Response: Thank you for the comments. We have revised the title accordingly, which now reads as:
Obesity-associated NAFLD coexists with a chronic inflammatory kidney condition that is partially mitigated by short-term oral metformin treatment.
2. Abstract, lines 19–21: The phrase "normal diet to induce Ob-NAFLD" may be confusing, as a "normal" diet, by definition, should not induce NAFLD. Although this is clarified later in the manuscript, the wording in the abstract may not be immediately clear to the reader. Consider removing the word "normal", or specifying the actual name of the diet (e.g., AIN-93G) to enhance clarity.
Response: The lines in the abstract are revised as:
…”Five-week-old female Zucker rats (obese fa/fa and lean Fa/Fa) were fed an AIN-93 G diet for 8 weeks to induce Ob-NAFLD, then fed the diet with Metformin for 10 weeks. Kidneys were collected for histopathological and biochemical analyses.” …
3. Given that the authors’ main hypothesis relies on the combined effect of obesity and NAFLD on the development of kidney disease, it is essential to clearly demonstrate that the animals used in this study were both obese and had NAFLD. While the use of a genetic obesity model supports the presence of obesity, the manuscript does not provide sufficient evidence that the animals developed NAFLD, neither it is specified in which degree or severity. It appears that these data may have been reported in previous publications. Using the same animals is, of course, entirely acceptable. However, in its current form, the manuscript does not clearly state whether the rats used in this study are the same as those described in the cited references. The authors should explicitly clarify this point in the text and provide the relevant references to confirm that the animals analyzed in the present study met both conditions.
Response: We apologize for not making this clearer earlier. We have revised these sentences under the Materials and Methods section (section 2.2 Experimental design) to bring clarity. The revised sentence now reads as:
…..All rats were fed an AIN-93 G diet from Envigo for 8 weeks to induce obesity and NAFLD (Ob-NAFLD). [15, 19, 20] ….
Similarly, we have revised the manuscript with a brief discussion and citation of prior published work in the Results section to bring clarity. The revised sentences now read as:
Previous studies demonstrated that female Zucker rats (fa/fa) that are fed with AIN-93 G diet produce obesity when compared to the female Zucker rats (Fa/Fa) [15, 19, 20]. The mean +/- SD body weight (g) of lean and obese rats at the beginning of the experiment was as follows: lean, 98 ± 7 g and obese, 155 ± 17g [15]. After 10 weeks, the body weight of the rats was as follows: lean, 269 ± 26 g, and obese, 590 ± 41 [15]. The obese rats displayed significantly altered serum metabolites and developed NAFLD as indicated by increased liver weight and steatosis, when compared to the lean rats [15, 19, 20]. Furthermore, the obese rats show increased liver weight and impaired liver function when compared to the lean rats [15], suggesting the obesity-associated NAFLD (Ob-NAFLD). In this study, we sought to determine if Ob-NAFLD has any consequences on kidney injury. To investigate this, we used the same rats as described in [15, 19, 20]. Rat kidneys were isolated and processed for biochemical and histopathological analyses. ….
….Previous studies utilizing the Ob-NAFLD rodent model demonstrated that the short-term metformin treatment (10 weeks) did not significantly affect body weight (lean rats: control, 269 ± 26 g; plus metformin, 278 ± 17 g; obese rats: obese, 590 ± 41, plus metformin, 573 ± 48 g) [15]. Although the metformin treatment did not alter the serum liver enzymes in Ob-NAFLD rats, it partially ameliorated liver histological changes (steatosis score) and improved serum metabolic parameters[15, 19, 20]. ……
4. Closely related to the previous comment, I find that the manuscript lacks some important biometrical and biochemical data for the rats, which are particularly relevant to assess the degree of obesity and the severity of NAFLD the rats developed, such as initial and final body weight, food intake, glucose status, etc. If these data have already been published elsewhere, please clearly provide the corresponding references so that readers can consult them. Additionally, a brief summary of the main health characteristics of the rats could be helpful (at the authors’ discretion, if the references are provided).
Response: Thank you for the suggestion. Please refer to our response above (regarding the published data) to your comment # 3. Unfortunately, our study design did not include an evaluation of glucose levels. This limitation is now discussed in our revised discussion section, which reads as:
… Similarly, our study design did not include the evaluation of blood glucose levels. …
5. Along the same lines, in the Discussion, lines 429-434: The authors suggest that variability in metformin intake due to ad libitum feeding may have contributed to the modest or inconsistent treatment effects. While this is a plausible explanation, it underscores a methodological limitation that should be more directly acknowledged. As noted in my previous comment, the manuscript does not include any biometric data—such as body weight evolution or food intake—which are essential when drug administration depends on dietary consumption. Even with ad libitum access, monitoring food intake per cage and correlating it with individual body weight would allow for a rough estimation of drug exposure per animal. This is a key point that the authors should clarify, as it directly impacts the interpretation of metformin's effectiveness. Including such data, or explicitly addressing its absence as a limitation, would considerably strengthen the validity of the conclusions.
Response: We agree. Please see our response above to your comment #3. The published reports (Reference # [15, 19, 20] ) show the evolution of body weight after feeding with the AIN-93 G diet in both lean and obese rats. However, a limitation in our approach is that we did not collect food intake data, and we realized that this is another weakness of our study design. Therefore, we have included this as a limitation in the revised Discussion section:
… Similarly, the administration of metformin was performed via the oral route with ad libitum diet access, and the monitoring of food intake was not included in the experimental design, which is another significant limitation of this study. …
6. I am not entirely sure about the statistical approach used. Why was a one-way ANOVA chosen? Which factor was considered for the analysis? It seems that there might actually be two relevant factors—rat strain and metformin treatment—so a two-way ANOVA could be more appropriate. Additionally, ANOVA (whether one-way or two-way) does not perform pairwise comparisons by itself. In the text, the authors state that t-tests were used for this purpose, but this is not reflected in the figure legends or results. Finally, although the answer is likely yes, did you verify whether the data meet the assumptions for parametric tests? This should be clearly specified in the methods section.
Response: We agree that the two-way ANOVA is a better approach while comparing groups with two factors (Lean vs Obese with or without metformin). We revised our statistical analysis using a two-way ANOVA followed by Tukey’s post hoc analysis to pinpoint specific differences. We did verify that the data meet the assumptions for parametric tests. If the data did not pass the normal distribution, we used the Mann-Whitney U test (non-parametric). We have now revised the Statistical analysis section as follows:
Kidneys isolated from the rats (n = 6 per group), as published before [15], were considered in this study. Statistical analysis was performed using GraphPad Prism (Version 10), and the data are presented as mean ± SD. The Shapiro-Wilk normality test was performed to verify normal distribution of the data. We performed a two-way ANOVA for comparisons among multiple groups. Tukey’s post hoc analysis (alpha=0.05) was performed when normal distribution assumptions were met. For the skewed data, the Mann-Whitney U test was performed to compare the difference between the two groups. Statistical significance was defined as a P-value < 0.05. Graphs were generated using GraphPad Prism.
7. I kindly suggest that the data mentioned in the manuscript but not shown be included as supplementary material. This would enhance transparency and allow readers to fully assess the study’s conclusions.
Response: Thanks. New supplementary information showing Supplemental figures 1 and 2 are included along with the revised manuscript.
8. Regarding the organization of the manuscript, I do not believe that placing all the figures under a single section is the best choice. It would be much clearer and more reader-friendly to position each figure as close as possible to its first citation in the text. In any case, the separate figure section should be removed to improve the overall flow and readability of the manuscript.
Response: Revised accordingly. We have positioned each figure as close as possible to its first citation in the text of the Results section.
9. Line 152: Please add the corresponding references and clearly specify whether these are the same rats. As mentioned earlier, this point is critical to support the authors' hypothesis.
Response: Thank you for the suggestion. We have revised the statistical section accordingly and cited the published work. Please refer to our response to your comment #6.
10. Minor textual issues:
1. If I am correct, Table 1 lists only the primary antibodies used in the study, despite the title suggesting otherwise. The authors may want to revise the table title to accurately reflect its contents.
Response: Correct. We have revised the title of the table accurately. The title now reads as:
Table 1: Primary antibodies used in this study.
2. Line 142: Please correct the capitalization of "Reverse" to "reverse."
Response: Thanks. Revised accordingly.
3. Lines 245-246: Please, revise this sentence: “We evaluated the levels of gene expression for TNF-α, Ccl2, and CXCR2 in rat kidneys after Ob-NAFL with or without metformin treatment.”
Response: Thanks. Revised accordingly. It now reads as:
We evaluated the levels of gene expression for TNF-α, Ccl2, and CXCR2 in rat kidneys after Ob-NAFLD with or without metformin treatment.
4. Line 378: Please, define FSGS.
Response: Thanks. Revised accordingly. It now reads as:
Histological analyses revealed substantial damage in both the tubular and glomerular compartments, characterized by moderate tubular injury and FSGS (focal segmental glomerulosclerosis) in the majority of Ob-NAFLD rats. Interstitial fibrosis, a hallmark of chronic kidney injury, was also present.
Reviewer 2 Report
Comments and Suggestions for Authors
The paper by Sharma et al describes the effect of metformin on Obesity-NAFLD - CKD status in rats. The paper is overall well-written and scientifically sound. The experimental design is appropriate, and the results are clearly presented and interpreted.
I have two questions and one comment which may be found below:
- Why did authors consider only females, instead of both sexes, as biological sex is found to be important in conditions such as CKD?
- Why did authos use oneway ANOVA instead of twoway ANOVA? Considering their experimental design it would maybe be more appropriate and more informative to use twoway ANOVA to see the interaction effect of metformin and NAFLD on kidney parameters.
- Please enlarge the font size and overall dimensions of the graphical displays, as they are currently difficult to read.
Author Response
Comments from Reviewer 2 and Author’s Response
Comments and Suggestions for Authors
The paper by Sharma et al describes the effect of metformin on obesity-NAFLD-CKD status in rats. The paper is overall well-written and scientifically sound. The experimental design is appropriate, and the results are clearly presented and interpreted.
Response:
I have two questions and one comment, which may be found below:
1. Why did authors consider only females, instead of both sexes, as biological sex is found to be important in conditions such as CKD?
Response: We agree. The experimental design was performed based on the existing evidence that both male and female obese Zucker rats develop obesity and hepatic steatosis at comparable rates. Therefore, we included only female Zucker rats. However, we agree that this is a limitation of our studies because sex is an important biological factor in driving the CKD condition. Therefore, we revised the discussion section, which now reads as follows:
One of the limitations of this study is the exclusive use of female Zucker rats. While this may raise questions regarding sex-specific responses, existing evidence suggests that in the Ob-NAFLD model, both male and female obese Zucker rats develop obesity and he-patic steatosis at comparable rates, with no significant sex-related differences in the pro-gression of metabolic or histological features [19, 53]. Therefore, the choice of female ani-mals is unlikely to have introduced a substantial bias in the evolution of body weight and liver steatosis, including the key endpoints of the published studies [15]. Nonetheless, fu-ture studies could benefit from including both sexes to fully rule out any subtle, sex-specific variations in renal response to therapeutic interventions, particularly to the CKD condition. Similarly, our study design did not include the evaluation of blood glu-cose levels. Another limitation lies in the lack of functional renal assessments. While the study focused primarily on histological changes and molecular markers of kidney injury, we could not evaluate serological functional parameters of kidney health, which are criti-cal for assessing therapeutic efficacy in a translational context. Functional biomarkers such as glomerular filtration rate, creatinine clearance, blood urea nitrogen, or proteinuria would provide valuable insights into how molecular and histological improvements translate into preserved or improved renal function. Similarly, the administration of met-formin was performed via the oral route with ad libitum diet access, and the monitoring of food intake was not included in the experimental design, which is another significant limitation of this study.
2. Why did the authors use one-way ANOVA instead of two-way ANOVA? Considering their experimental design, it would maybe be more appropriate and more informative to use two-way ANOVA to see the interaction effect of metformin and NAFLD on kidney parameters.
Response: Thank you for your suggestions. We agree. Reviewer 1 has also brought this point. We have revised our statistical analysis approach using a two-way ANOVA with Tukey’s post-hoc analysis (parametric) and the Mann-Whitney U test (non-parametric). In addition, we performed a normality test of our variables and used non-parametric tests wherever the data was skewed. Please see our response to Reviewer 1’s Comment # 6 on this issue.
3. Please enlarge the font size and overall dimensions of the graphical displays, as they are currently difficult to read.
Response: We have revised it accordingly.
Round 2
Reviewer 1 Report
Comments and Suggestions for Authors
I appreciate the efforts of the authors to address all my comments. I feel quite satisfied with the corrections made. However, the fact that the authors did not record food intake or measure blood glucose levels (or any other parameter related to insulin and glucose metabolism) remains a critical issue, especially considering that they are testing the effects of an anti-diabetic drug added to the regular diet. Without data on food intake, it is very difficult, if not imposible, to determine the actual dose of metformin the rats received during the experiment. I encourage the authors to take this into account in the design of future experiments.
I also have one further comment regarding the statistical analysis. In the captions of Figures 2 and 6, the authors state that a two-way ANOVA was conducted. However, it appears that the results shown are not from the ANOVA itself, but rather from pairwise comparisons between groups, likely obtained through a post hoc Tukey test (as described in the Materials and Methods section). A two-way ANOVA typically identifies which factors are contributing to the differences and whether there is an interaction between the two factors being considered. In contrast, a post hoc test reveals which specific group differences are statistically significant. Please revise the figure captions accordingly.